# DiffGCN: Graph Convolutional Networks via Differential Operators and Algebraic Multigrid Pooling

**Moshe Eliasof**
Department of Computer Science
Ben-Gurion University of the Negev
Beer-Sheva, Israel
eliasof@post.bgu.ac.il

**Eran Treister**
Department of Computer Science
Ben-Gurion University of the Negev
Beer-Sheva, Israel
erant@cs.bgu.ac.il

## Abstract

Graph Convolutional Networks (GCNs) have shown to be effective in handling unordered data like point clouds and meshes. In this work we propose novel approaches for graph convolution, pooling and unpooling, inspired from finite differences and algebraic multigrid frameworks. We form a parameterized convolution kernel based on discretized differential operators, leveraging the graph mass, gradient and Laplacian. This way, the parameterization does not depend on the graph structure, only on the meaning of the network convolutions as differential operators. To allow hierarchical representations of the input, we propose pooling and unpooling operations that are based on algebraic multigrid methods, which are mainly used to solve partial differential equations on unstructured grids. To motivate and explain our method, we compare it to standard convolutional neural networks, and show their similarities and relations in the case of a regular grid. Our proposed method is demonstrated in various experiments like classification and part-segmentation, achieving on par or better than state of the art results. We also analyze the computational cost of our method compared to other GCNs.

## 1 Introduction

The emergence of deep learning and Convolutional Neural Networks (CNNs) [1, 2, 3] in recent years has had great impact on the community of computer vision and graphics [4, 5, 6, 7]. Over the past years, multiple works used standard CNNs to perform 3D related tasks on unordered data (e.g., point clouds and meshes), one of which is PointNet [8, 9], that operates directly on point clouds. Along with these works, another massively growing field is Graph Convolutional Networks (GCNs) [10], or Geometric Deep Learning, which suggests using graph convolutions for tasks related to three dimensional inputs, arising from either spectral theory [11, 12, 13] or spatial convolution [14, 15, 16, 17]. This makes the processing of unstructured data like point clouds, graphs and meshes more natural by operating directly in the underlying structure of the data.

In this work we aim to bridge the gap between ordered and unordered deep learning architectures, and to build on the foundation of standard CNNs in unordered data. To this end, we leverage the similarity between standard CNNs and partial differential equations (PDEs) [18], and propose a new approach to define convolution operators on graphs that are based on discretization of differential operators on unstructured grids. Specifically, we define a 3D convolution kernel which is based on discretized differential operators. We consider the mass (self-feature), gradient and Laplacian of the graph, and discretize them using a simple version of finite differences, similarly to the way that standard graph Laplacians are defined. Such differential operators form a subspace which spans standard convolution

kernels on structured grids. Leveraging such operators for unstructured grids leads to an abstract parameterization of the convolution operation, which is independent of the specific graph geometry.

Our second contribution involves unstructured pooling and unpooling operators, which together with the convolution, are among the main building blocks of CNNs. To this end, and further motivated by the PDE interpretation of CNNs, we utilize multigrid methods which are among the most efficient numerical solvers for PDEs. Such methods use a hierarchy of smaller and smaller grids to represent the PDE on various scales. Specifically, algebraic multigrid (AMG) approaches [19, 20] are mostly used to solve PDEs on unstructured grids by forming the same hierarchy of problems using coarsening and upsampling operators. Using these building blocks of AMG, we propose novel pooling and unpooling operations for GCNs. Our operators are based on the Galerkin coarsening operator of aggregation-based AMG [21, 22], performing pure aggregation for pooling and smoothed aggregation as the unpooling operator. The advantage of having pooling capability, as seen both in traditional CNNs and GCNs [4, 23, 5, 6] are the enlargement of the receptive field of the neurons, and reduced computational cost (in terms of floating operations), allowing for wider and deeper networks.

In what follows, we elaborate on existing unordered data methods in Section 2, and present our method in Section 3. We discuss the similarity between traditional CNNs and our proposed GCN, and motivate the use of differential operators as a parameterization to a convolution kernel in Section 3.3. Furthermore, we compare the computational cost of our method compared to other message-passing, spatially based GCNs in Section 3.5 . To validate our model, we perform experiments on point cloud classification and segmentation tasks on various datasets in Section 4. Finally, we study the importance and contribution of the different terms in the parameterization to the performance of our method in Section 4.3.

## 2    Related work

Unordered data come in many forms and structures – from meshes and point clouds that describe 3D objects to social network graphs. For 3D related data, a natural choice would be to voxelize the support of the data, as in [24]. Clearly, such approach comes at a high computational cost, while causing degradation of the data. Other methods suggest to operate directly on the data - whether it is a point cloud [8, 9, 25] or a graph [13, 15, 11, 12, 26].

Recent works like [15, 17] assumed a system of local coordinates centered around each vertex. These methods propose to assign weights to geometric neighborhoods around the vertices, in addition to the filter weights. Masci et al. [15] proposed assigning fixed Gaussian mixture weight for those neighborhoods, and [17] goes a step further and learns the parameters of the Gaussians. These methods require high computational costs, due to the computation of exponential terms (particularly at inference time) as well as the overhead of additional learnable parameters.

Later, it was shown in [26] that adopting GCNs for point-cloud related tasks can be highly beneficial, since the learned features of the graph vertices in different layers of the network can induce dynamic graphs which reveal their underlying correlations. We follow the trend of employing GCNs for point-cloud related tasks like shape classification and segmentation. Specifically, we choose to work with spatial GCNs since they are most similar to standard structured CNNs. However, compared to other works like DGCNN [27] and MPNN [16] which can be interpreted as non-directed discretized gradient operators, we introduce directed gradients, as well as the addition of the Laplacian term of the graph. The Laplacian is the key ingredient in spectral-based methods like [13, 12], but was not used in spatial GCNs where the vertices have a geometric meaning, to the best of our knowledge.

Unlike traditional structured CNNs, where the pooling and unpooling operations are trivial, these operations are more debatable in unordered methods, due to the lack of order or the metric between points. Works like PointNet++ [9] proposed using Furthest Point Sampling technique in order to choose remaining points in coarsened versions of the inputs. Other works proposed utilizing $\ell_2$ norm of the features to determine which elements of the graph are to be removed in subsequent layers of the network [6, 28]. Recent works like DiffPool [29] proposed learning a dense assignment matrix to produce coarsened version of an initial graph. However, learning a dense matrix is of quadratic computational cost in the number of vertices and does not scale well for large scale point-clouds. Also, DiffPool is constrained to fixed graph sizes, while our method is agnostic to the size of the input.

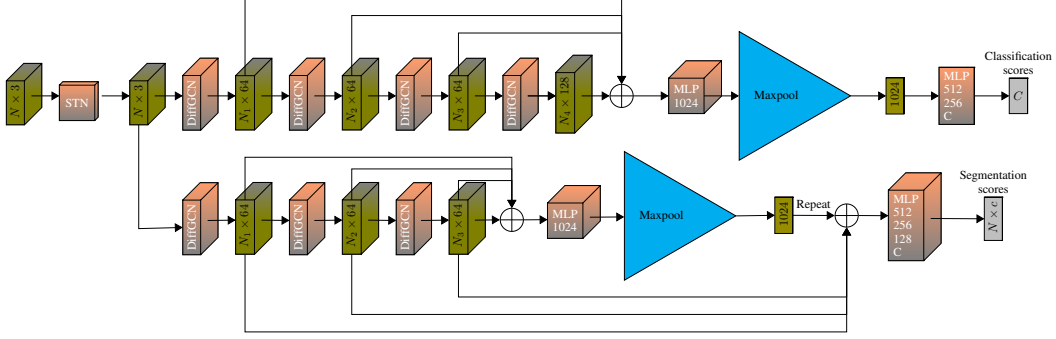

Figure 1: Our architectures for classification (upper part) and segmentation (lower part). STN denotes a spatial transformer module [8]. Channel-wise concatenation is denoted by $\oplus$. $N_i$ denotes the number of vertices after the $i$-th DiffGCN block.

We focus on the employment of AMG as a concept to define our pooling and unpooling operations. We use classical aggregation AMG [22], which is suitable for unstructured grids, similarly to works that incorporated geometric multigrid concepts into structured grids CNNs [5, 30]. On a different note, the recent work [31] showed another connection between AMG and GCNs, and proposed using GCNs for learning sparse AMG prolongation matrices to solve weighted diffusion problems on unstructured grids.

## 3    Method

We propose to parameterize the graph convolutional kernel according to discretized differential operators defined on a graph. Therefore, we call our convolution DiffGCN. To have a complete set of neural network building blocks, we also propose an AMG inspired pooling and unpooling operators to enlarge the receptive fields of the neurons, and to allow for wider and deeper networks.

### 3.1    Convolution kernels via differential operators

To define the convolution kernels in the simplest manner, we use finite differences, which is a simple and widely used approach for numerical discretization of differential operators. Alternatives, such as finite element or finite volume schemes may also be suitable for the task, but are more complicated to implement in existing deep learning frameworks. Using finite differences, the first and second order derivatives are approximated as:

$$\frac{\partial f(x)}{\partial x} \approx \frac{f(x+h) - f(x-h)}{2h}, \quad \frac{\partial^2 f(x)}{\partial x^2} \approx \frac{f(x+h) - 2f(x) + f(x-h)}{h^2}. \tag{1}$$

We harness these simple operators to estimate the gradient and Laplacian of the unstructured feature maps defined on a graph.

Given an undirected graph $\mathcal{G} = (V, E)$ where $V, E$ denote the vertices and edges of the graph, respectively, we propose a formulation of the convolution kernel as follows:

$$conv(\mathcal{G}, \Theta) \approx \theta_1 I + \theta_2 \frac{\partial}{\partial x} + \theta_3 \frac{\partial^2}{\partial x^2} + \theta_4 \frac{\partial}{\partial y} + \theta_5 \frac{\partial^2}{\partial y^2} + \theta_6 \frac{\partial}{\partial z} + \theta_7 \frac{\partial^2}{\partial z^2}. \tag{2}$$

This gives a 7-point convolution kernel which consists of the mass, gradient and Laplacian of the signal defined over the graph.

We now formulate the operators in (2) mathematically. We first define that the features of the GCN are located in the vertices $v_i \in V$ of the graph, similarly to a nodal discretization. For each node we have $c_{in}$ features (input channels). We start with the definition of the gradient (in $x, y, z$), which according to (1) is defined on the middle of an edge $e_{ij} \in E$ connecting the pair $v_i$ and $v_j$. Since the edge direction may not be aligned with a specific axis, we project the derivative along the edge onto the axes $x, y, z$. For example,

$$(\partial_{\mathcal{G}}^x f)_{ij} = \frac{\partial f}{\partial x}(e_{ij}) = \frac{f_{v_i} - f_{v_j}}{dist(v_i, v_j)}(x(v_i) - x(v_j)). \tag{3}$$

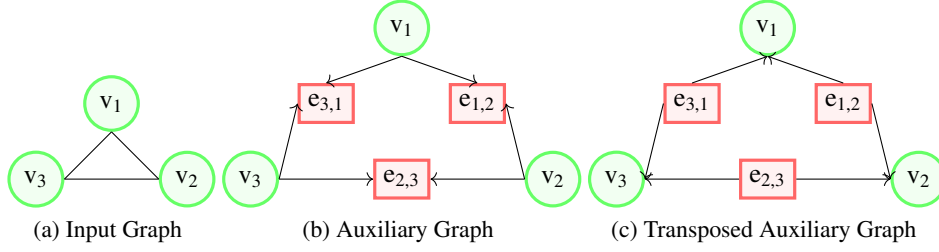

(a) Input Graph      (b) Auxiliary Graph      (c) Transposed Auxiliary Graph

Figure 2: An example of a graph and its auxiliary and transposed auxiliary graphs.

$x(v_i)$ is the $x$-coordinate of vertex $v_i$, and $f_{v_i} \in \mathbb{R}^{c_{in}}$ is the feature vector of size $c_{in}$ defined on vertex $v_i$. $dist(v_i, v_j)$ is the Euclidean distance between $v_i$ and $v_j$. Given this approach, we define the gradient matrix of the graph by stacking the projected differential operator in each of the axes $x, y, z$:

$$\nabla_{\mathcal{G}} = \begin{bmatrix} \partial^x_{\mathcal{G}} \\ \partial^y_{\mathcal{G}} \\ \partial^z_{\mathcal{G}} \end{bmatrix} : |V| \cdot c_{in} \to 3 \cdot |E| \cdot c_{in} \tag{4}$$

This gradient operates on the vertex space and its output size is 3 times the edge space of the graph, for the $x, y, z$ directions. To gather the gradients back to the vertex space we form an edge-averaging operator

$$A : 3 \cdot |E| \cdot c_{in} \to 3 \cdot |V| \cdot c_{in}, \quad (Af)_i = \frac{1}{|\mathcal{N}_e(v_i)|} \sum_{j \in \mathcal{N}_e(v_i)} f_{e_{ij}} \tag{5}$$

where $\mathcal{N}_e(v_i) = \{j : e_{ij} \in E\}$ is the set of edges associated with vertex $v_i$. The function $f$ in (5) is a feature map tensor defined on the edges of the graph. The three different derivatives in (4) are treated as three different features for each edge.

In a similar fashion, the Laplacian of the graph with respect to each axis $x, y, z$ is computed in two steps. The first step is the gradient in equation (4). Then, we apply the following transposed first-order derivative operators to obtain the second derivatives back on the vertices:

$$\begin{bmatrix} (\partial^x_{\mathcal{G}})^T & 0 & 0 \\ 0 & (\partial^y_{\mathcal{G}})^T & 0 \\ 0 & 0 & (\partial^z_{\mathcal{G}})^T \end{bmatrix} : 3 \cdot |E| \cdot c_{in} \to 3 \cdot |V| \cdot c_{in}. \tag{6}$$

The transposed gradient if often used to discretize the divergence operator, and the resulting Laplacian is the divergence of the gradient. Here, however, we do not sum the derivatives (the Laplacian is the sum of all second derivatives) so that the second derivative in each axis ends up as a separate feature on a vertex, so it is weighted in our convolution operator in (2). This construction is similar to the way graph Laplacians and finite element Laplacians are defined on graphs or unstructured grids.

**Implementation using PyTorch-Geometric**    To obtain such functionality while using common GCN-designated software [32] and concepts [14, 16], we define a directed *auxiliary* graph denoted by $\mathcal{G}' = (V', E')$, where in addition to the original set of vertices $V$, we have new dummy vertices, representing the mid-edge locations $e_{ij} \in E$. Then, we define the connectivity of $\mathcal{G}'$ such that each vertex $v_i \in V$ has a direct connection to the mid-edge location $e_{ij}$ as in Fig. 2. More explicitly:

$$V' = V \cup E \quad , \quad E' = \{(v_i, e_{ij}), (v_j, e_{ij}) \mid e_{ij} \in E\}. \tag{7}$$

We also use the transposed graph, which is an edge flipped version of $\mathcal{G}'$, also demonstrated in Fig. 2.

Given these two graphs, we are able to obtain the gradient and Laplacian terms of the signal defined over the graph via mean aggregation of message passing scheme [16, 14], where we perform two stages of the latter. First, we use the auxiliary graph $\mathcal{G}'$ to send the following message for each $(v_i, e_{ij}) \in E'$:

$$msg_{Grad}(v_i \to e_{ij}, f_{v_i}) = \frac{f_{v_i}}{2 \cdot dist(v_i, e_{ij})} \left( \begin{bmatrix} x(v_i) \\ y(v_i) \\ z(v_i) \end{bmatrix} - \begin{bmatrix} x(e_{ij}) \\ y(e_{ij}) \\ z(e_{ij}) \end{bmatrix} \right) \in \mathbb{R}^{3 \cdot c_{in}}. \tag{8}$$

Here, each vertex gets two messages, and due to the subtraction of vertex locations in the message followed by a sum aggregation

$$g_{e_{ij}} = msg_{Grad}(v_i \rightarrow e_{ij}, f_{v_i}) + msg_{Grad}(v_j \rightarrow e_{ij}, f_{v_j}) \tag{9}$$

the discretized gradient in (3)-(4) is obtained on the edges. Following this, we return two messages over the transposed graph $\mathcal{G}'$, returning both the gradient and the Laplacian of the graph on the original vertex space $V$. The first part of the message returns the gradient terms from the edges to the vertices simply by sending the identity message followed by mean aggregation:

$$Grad_{\mathcal{G}}(v_i) = \frac{1}{|\mathcal{N}_i|} \sum_{j \in \mathcal{N}_i} g_{e_{ij}}. \tag{10}$$

This concludes Eq. (5). The second part of the message differentiates the edge gradients to obtain the Laplacian back on the original vertices:

$$msg_{EdgeLap}(e_{ij} \rightarrow v_i, g_{e_{ij}}) = \frac{g_{e_{ij}}}{2 \cdot dist(v_i, e_{ij})} \left( \begin{bmatrix} x(e_{ij}) \\ y(e_{ij}) \\ z(e_{ij}) \end{bmatrix} - \begin{bmatrix} x(v_i) \\ y(v_i) \\ z(v_i) \end{bmatrix} \right) \in \mathbb{R}^{3 \cdot c_{in}}. \tag{11}$$

Then, we obtain the features described in Eq. (6) by performing mean aggregation:

$$Lap_{\mathcal{G}}(v_i) = \frac{1}{|\mathcal{N}_i|} \sum_{j \in \mathcal{N}_i} msg_{EdgeLap}(e_{ij}, v_i). \tag{12}$$

Finally, we concatenate the mass, gradient and Laplacian to obtain the *differential operators features*:

$$\hat{f}_{v_i} = f_{v_i} \oplus Grad_{\mathcal{G}}(v_i) \oplus Lap_{\mathcal{G}}(v_i) \in \mathbb{R}^{7 \cdot c_{in}}, \tag{13}$$

where $\oplus$ denotes channel-wise concatenation. Finally, we apply a multi-layer perceptron (MLP)—a $c_{out} \times 7 \cdot c_{in}$ point-wise convolution followed by batch normalization and ReLU— to the features in Eq. (13).

The implementation above follows the mathematical formulation step by step, but it requires to explicitly construct the auxiliary graph in Fig. 2. An equivalent and more efficient way to implement our method, which is only implicitly based on those auxiliary graphs, is to construct a message that contains $6 \cdot c_{in}$ features by combining Eq. (9) and (11) in a single message followed by mean aggregation as in Eq. (12) and concatenation of the self feature, resulting in a feature $\hat{f}_{v_i} \in \mathbb{R}^{7 \cdot c_{in}}$.

## 3.2 Algebraic multigrid pooling and unpooling

An effective pooling operation is important to faithfully represent coarsened versions of the graph. We propose to use AMG methods [22, 20], namely, the Galerkin coarsening which we explain now.

In AMG methods the coarse graph vertices are typically chosen either as a subset of the fine graph vertices (dubbed "C-points") or as clusters of vertices called aggregates. We use the latter approach, and apply the Graclus clustering [33] to form the aggregates. Let $\{\mathcal{C}_J\}_{J=1}^{|V_{coarse}|}$ be the aggregates, each corresponds to a vertex in the coarse graph. Then, we define the restriction (pooling) operator:

$$R_{J,i} = \begin{cases} 1 & i \in \mathcal{C}_J \\ 0 & otherwise \end{cases}. \tag{14}$$

Given a features matrix $X$ and an adjacency matrix $A$, their coarsened counterparts are defined via the Galerkin coarsening:

$$X^{coarse} = R^T X, \quad A^{coarse} = R^T A R \in \mathbb{R}^{|V_{coarse}| \times |V_{coarse}|}. \tag{15}$$

To perform the unpooling operator, also called prolongation, we may use the transpose of the restriction operator (14). However, when unpooling with an aggregation matrix, we get piece-wise constant feature maps, which are undesired. To have a smoother unpooling operator, we propose to allow the prolongation of soft clustering via *smoothed* aggregation [22] as follows:

$$P = (I - (D)^{-1}L)R^T \in \mathbb{R}^{|V| \times |V_{coarse}|} \tag{16}$$

Where $I, D, L$ are the identity matrix, degree and Laplacian matrix of the layer, respectively. To unpool from a coarsened version of the graph, we apply the corresponding prolongation operator at each level, until we reach the initial problem resolution.

### 3.3 Similarity between DiffGCN and standard CNN operators for structured grids

A standard CNN is based on learning weights of convolutional filters. The work [18] showed that the 2D convolution kernel can be represented as a linear combination of finite difference differential operators. These classical differential operators are obtained using our definitions in Eq. (3)-(6), in the case of a structured regular graph. In 2D (without the $z$ axis), Eq. (2) will result in a 5-point stencil represented as

$$\theta_1 \begin{bmatrix} 0 & 0 & 0 \\ 0 & 1 & 0 \\ 0 & 0 & 0 \end{bmatrix} + \theta_2 \begin{bmatrix} 0 & 0 & 0 \\ -1 & 0 & 1 \\ 0 & 0 & 0 \end{bmatrix} + \theta_3 \begin{bmatrix} 0 & 0 & 0 \\ 1 & -2 & 1 \\ 0 & 0 & 0 \end{bmatrix} + \theta_4 \begin{bmatrix} 0 & 1 & 0 \\ 0 & 0 & 0 \\ 0 & -1 & 0 \end{bmatrix} + \theta_5 \begin{bmatrix} 0 & 1 & 0 \\ 0 & -2 & 0 \\ 0 & 1 & 0 \end{bmatrix} \quad (17)$$

The Laplacian, together with the mass term allow the network to obtain low-pass filters, which are highly important to average out noise, and to prevent aliasing when downsampling the feature-maps. Gradient based methods like [26] can only approximate the Laplacian term via multiple convolutions, leading to redundant computations. Furthermore, the work of [34] showed that the popular $3 \times 3$ convolution kernel can be replaced by this 5 point stencil without losing much accuracy. When extending this to 3D, the common $3 \times 3 \times 3$ kernel includes 27 weights, and the lighter version in (2) ends in a star-shaped stencil using 7 weights only, which is a significant reduction from 27. We refer the interested reader to [35, 36, 37, 38, 39, 40] for a more rigorous study of the connection between ODEs, PDEs and CNNs.

### 3.4 DiffGCN architectures

We show the architectures used in this work in Fig. 1. We define a DiffGCN block which consists of two DiffGCN convolutions, with a shortcut connection, as in ResNet [23] for better convergence and stability. Pooling is performed before the first convolution in each block, besides the first opening layer. We use concatenating skip-connections to fuse feature maps from shallow and deep layers. Before this concatenation, unpooling is performed to resize the point-cloud to its original dimensions.

### 3.5 Computational cost of DiffGCN

Typically, spatial GCNs like [14, 16, 26, 15] employ the convolutions $K$ times per vertex, where $K$ is the neighborhood size. More explicitly, a typical convolution can be written as

$$x'_i = \underset{j \in \mathcal{N}_i}{\square} h_\Theta(f(x_i, x_j)), \quad (18)$$

where $\mathcal{N}_i$ is the set of neighbors of vertex $v_i \in V$, $\square$ is a permutation invariant aggregation operator like max or sum and $h_\Theta$ is an MLP [8] parameterized by the set of weights $\Theta$. $f$ is a function that is dependent on a vertex and its neighbors. For instance, in DGCNN [26] $f(x_i, x_j) = x_i \oplus (x_i - x_j)$. By design, our convolution operation first gathers the required differential terms, and then feeds their channel-wise concatenation through a MLP. That is, our convolution can be written as

$$x'_i = h_\Theta(\underset{j \in \mathcal{N}_i}{\square} g(x_i, x_j)), \quad (19)$$

where $g$ is a function that constructs the desired differential operator terms. Thus, we reduce the feed-forward pass of our convolution by an order of $K$, which decreases the number of FLOPs required in our convolution. In other words, the MLP operation in our convolution is independent of the number of neighbors $K$, since we aggregate the neighborhood features prior to the MLP. If $s$ is the stencil size (e.g.,DGCNN [26] uses $s = 2$, while ours is $s = 7$), $N$ is the input size, and $c_{in}$, $c_{out}$ are the number of input and output channels, respectively, then the number of floating point operations of a method defined via Eq. (18) is $\mathcal{O}(s \times N \times K \times c_{in} \times c_{out})$, while the cost of our method in Eq. (19) reduces to $\mathcal{O}(s \times N \times c_{in} \times c_{out})$. In Table 1 we report the required FLOPs and latency for various convolutions with $1,024$ points input and $c_{in} = 64$, $c_{out} = 128$. For VoxNet [24] we use a $3 \times 3 \times 3$ kernel and $12 \times 12 \times 12$ input. For PointCNN, DGCNN and ours, we set the neighborhood size $K = 10$.

## 4 Experiments

To demonstrate the effectiveness of our framework, we conducted three experiments on three different datasets - classification (ModelNet40 [41]), part segmentation (ShapeNet Parts [42]) and semantic

Table 1: A comparison of single convolution FLOPs and latency

|  | VoxNet [24] | PointNet (MLP) [8] | PointCNN [25] | DGCNN [26] | DiffGCN (ours) |
|---|---|---|---|---|---|
| FLOPs[M] | 382.2 | 8.4 | 122.6 | 167.8 | 61.3 |
| LATENCY [ms] | 224 | 21 | - | 121 | 58 |

Table 2: Classification results on ModelNet40.

| Method | Mean Class Accuracy | Overall Accuracy |
|---|---|---|
| 3DShapeNets [41] | 77.3 | 84.7 |
| VoxNet [24] | 83.0 | 85.9 |
| Subvolume [47] | 86.0 | 89.2 |
| VRN (single view) [48] | 88.98 | — |
| VRN (multiple views) [48] | 91.33 | — |
| ECC [14] | 82.3 | 87.4 |
| PointNet [8] | 86.0 | 89.2 |
| PointNet++ [9] | — | 90.7 |
| Kd-net [49] | — | 90.7 |
| PCNN [50] | — | 92.3 |
| PointCNN [25] | 88.1 | 92.2 |
| KCNet [51] | — | 91.0 |
| DGCNN [26] (K=20) | 90.2 | 92.9 |
| DGCNN [26] (K=10) | 88.9 | 91.4 |
| LDGCNN [52] | 90.3 | 92.9 |
| Ours (K=20) | 90.4 | 93.5 |
| Ours (K=20, pooling) | 90.7 | 93.9 |

segmentation (S3DIS [43]). We also report an ablation study to obtain a deeper understanding of our framework. In all the experiments, we start from a point-cloud, and at each DiffGCN block we construct a K-nearest-neighbor graph according to the features of the points. As noted in [14], spectral methods generally lead to inferior results than spatial methods - thus we omit them in our experimental evaluations.

We implement our work using the PyTorch [44] and PyTorch Geometric [32] libraries. We use the networks shown in Fig. 1. For the semantic segmentation task on S3DIS we do not use a spatial transformer. Throughout all the experiments we use ADAM optimizer [45] with initial learning rate of 0.001. We run our experiments using NVIDIA Titan RTX with a batch size 20. Our loss function is the cross-entropy loss for classification and focal-loss [46] for the segmentation tasks.

## 4.1   Classification results

For the classification task we use ModelNet-40 dataset [41] which includes 12,311 CAD meshes across 40 different categories. The data split is as follows: 9,843 for training and 2,468 for testing. Our training scheme is similar to the one proposed in PointNet [8], in which we rescale each mesh to a unit cube, and then we sample 1,024 random points from each mesh at each epoch. We also use random scaling between 0.8 to 1.2 and add random rotations to the generated point cloud. We report our results with $K = 20$, with and without pooling. The results of our method are summarized in Table 2. We obtained higher accuracy than [26, 51, 52] which also use GCNs for this task. We suggest that the difference stems mainly from the addition of the Laplacian term to our convolution, and the contribution of the pooling module. Note, the work HGNN [53] which is based on hyper-graphs, using features that are of size 4,096 (and not only 3), extracted from MVCNN [54] and GVCNN [55], therefore, we do not include it in Table 2.

## 4.2   Segmentation results

We test our method on two different segmentation datasets - Shapenet part segmentation [42] and Stanford Large-Scale 3D Indoor Spaces Dataset (S3DIS) [43]. We use the lower part network in Fig.

Table 3: ShapeNet part segmentation. Results shown in mean intersection over union metric.

| | Aero | Bag | Cap | Car | Chair | Ear phone | Guitar | Knife | Lamp | Laptop | Motor | Mug | Pistol | Rocket | Skate board | Table | Mean |
|---|---|---|---|---|---|---|---|---|---|---|---|---|---|---|---|---|---|
| Shapes | 2690 | 76 | 55 | 898 | 3578 | 69 | 787 | 392 | 1547 | 451 | 202 | 184 | 283 | 66 | 152 | 5271 | |
| PointNet [8] | 83.4 | 78.7 | 82.5 | 74.9 | 89.6 | 73.0 | 91.5 | 85.9 | 80.8 | 95.3 | 65.2 | 93.0 | 81.2 | 57.9 | 72.8 | 80.6 | 83.7 |
| PointNet++ [9] | 82.4 | 79.0 | 87.7 | 77.3 | 90.8 | 71.8 | 91.0 | 85.9 | 83.7 | 95.3 | 71.6 | 94.1 | 81.3 | 58.7 | 76.4 | 82.6 | 85.1 |
| KD-Net [49] | 80.1 | 74.6 | 74.3 | 70.3 | 88.6 | 73.5 | 90.2 | 87.2 | 81.0 | 94.9 | 57.4 | 86.7 | 78.1 | 51.8 | 69.9 | 80.3 | 82.3 |
| LocalFeature [57] | 86.1 | 73.0 | 54.9 | 77.4 | 88.8 | 55.0 | 90.6 | 86.5 | 75.2 | 96.1 | 57.3 | 91.7 | 83.1 | 53.9 | 72.5 | 83.8 | 84.3 |
| PCNN [50] | 82.4 | 80.1 | 85.5 | 79.5 | 90.8 | 73.2 | 91.3 | 86.0 | 85.0 | 95.7 | 73.2 | 94.8 | 83.3 | 51.0 | 75.0 | 81.8 | 85.1 |
| PointCNN [25] | 84.1 | 86.45 | 86.0 | 80.8 | 90.6 | 79.7 | 92.3 | 88.4 | 85.3 | 96.1 | 77.2 | 95.3 | 84.2 | 64.2 | 80.0 | 83.0 | 86.1 |
| KCNet [51] | 82.8 | 81.5 | 86.4 | 77.6 | 90.3 | 76.8 | 91.0 | 87.2 | 84.5 | 95.5 | 69.2 | 94.4 | 81.6 | 60.1 | 75.2 | 81.3 | 84.7 |
| DGCNN [26] | 84.0 | 83.4 | 86.7 | 77.8 | 90.6 | 74.7 | 91.2 | 87.5 | 82.8 | 95.7 | 66.3 | 94.9 | 81.1 | 63.5 | 74.5 | 82.6 | 85.2 |
| LDGCNN [52] | 84.0 | 83.0 | 84.9 | 78.4 | 90.6 | 74.4 | 91.0 | 88.1 | 83.4 | 95.8 | 67.4 | 94.9 | 82.3 | 59.2 | 76.0 | 81.9 | 85.1 |
| Ours | 85.1 | 83.1 | 87.2 | 80.9 | 90.9 | 79.8 | 92.1 | 87.8 | 85.2 | 96.3 | 76.6 | 95.8 | 84.2 | 61.1 | 77.5 | 83.6 | 86.4 |
| Ours (pooling) | 85.1 | 83.7 | 88.0 | 80.3 | 91.1 | 80.0 | 92.0 | 87.5 | 85.3 | 95.8 | 76.0 | 95.9 | 83.8 | 65.6 | 77.3 | 83.7 | 86.4 |

Table 4: Semantic segmentation results on Stanford Large-Scale 3D Indoor Spaces Dataset (S3DIS). Results shown in mean intersection over union metric.

| Method | mIoU | Overall Accuracy |
|---|---|---|
| PointNet (baseline) [8] | 20.1 | 53.2 |
| PointNet [8] | 47.6 | 78.5 |
| MS + CU(2) [59] | 47.8 | 79.2 |
| G + RCU [59] | 49.7 | 81.1 |
| PointCNN [25] | 65.4 | — |
| DGCNN [26] | 56.1 | 84.1 |
| DCM-Net [58] | 64.0 | — |
| Ours | 61.1 | 85.3 |
| Ours (pooling) | 61.4 | 85.5 |

1, with $K = 20, 10, 5$ in each of the DiffGCN blocks, respectively. For Shapenet part segmentation dataset, our objective is to classify each point in a point-cloud to its correct part category. There are 16,881 3D shapes across 16 different categories, with a total of 50 part annotation classes, where each shape is annotated with 2-6 parts. We sample 2,048 points from each shape and use the training, validation and testing split in [56]. The results are reported in Table 3. Our method achieves the highest mIoU out of all the considered networks.

The Stanford Large-Scale 3D Indoor Spaces Dataset (S3DIS) contains 3D scans of 272 room from 6 different areas. Each point is annotated with one of 13 semantic classes. We adopt the pre-processing steps of splitting each room into $1m \times 1m$ blocks with random 4,096 points at the training phase, and all points during, where each point represented by a 9D vector (XYZ, RGB, normalized spatial coordinates). We follow the training, validation and testing split from [8]. We follow the 6-fold protocol for training and testing as in [43], and report the results of this experiment in Table 4. We obtained higher accuracy than the popular point-based network PointNet [8] as well as the graph based network DGCNN [26]. Note that [25] uses different pre-processing steps. Namely, the blocks were of $1.3m \times 1.3m$, where the added $0.3m$ on each dimensions is used for location context, and is not part of the objective at each block. In addition, we compare our work with a recent work, DCM-Net [58], which differs from our method by its approach of combining geodesic and Euclidean data, decoupling the data by utilizing parallel networks.

## 4.3 Ablation study

We measure the contribution of each component of our model, as well as different combinations of them, on classification with ModelNet40. Our results read that as expected, using each component on its own (e.g., mass term only) reduces accuracy. However, by combining the different terms - accuracy increases. We found that using the mass and Laplacian term is more beneficial than the mass and gradient term. This shows the representation power of the Laplacian operator which is widely used in classical computer graphics and vision [60, 61, 62]. That is in addition to spectral-based GCNs which are parameterized by polynomials of the graph Laplacian [13, 12, 63].

Table 5: Ablation study results on ModelNet40.

| Variation | Mean Class Accuracy | Overall Accuracy |
|---|---|---|
| Mass+Grad+Lap (K=10) | 89.1 | 92.7 |
| Mass+Grad+Lap (K=10, w.pooling) | 89.5 | 93.1 |
| Mass+Grad+Lap (K=5) | 88.7 | 92.1 |
| Mass+Grad+Lap (K=5, w.pooling) | 88.9 | 92.3 |
| Mass Only (K=20) | 85.4 | 88.2 |
| Grad Only (K=20) | 79.9 | 85.0 |
| Lap Only (K=20) | 79.2 | 83.2 |
| Mass + Grad (K=20) | 88.3 | 91.0 |
| Mass + Lap (K=20) | 88.6 | 91.9 |

In addition, we experiment with different number of neighbors, with and without pooling, reading slight reduction in performance, but with less FLOPs and memory requirements. We note that the pooling operations lead to better performance since they enlarge the receptive fields of the neurons.

## 5 Conclusion

We presented a novel graph convolution kernel based on discretized differential operators, which together with our AMG pooling and unpooling operators form the most important components of a CNN. Our GCN network shows on par or better performance than current state-of-the-art GCNs. We also draw an analogy between standard structured CNNs and our method, and the reduced cost of ours compared to other GCNs.

## Broader Impact

The method we propose can be used for additional tasks in which the data have a geometric meaning. For instance, data sourced from geographic information systems (GIS) can be used for prediction of elections results [64]. Thus, it may have an impact on other fields. In addition, our method is lighter than other GCNs, which can be beneficial for power and time consumption. We are not aware of an ethical problem or negative societal consequences.

## Acknowledgments and Disclosure of Funding

The research reported in this paper was supported by the Israel Innovation Authority through Avatar consortium, and by grant no. 2018209 from the United States - Israel Binational Science Foundation (BSF), Jerusalem, Israel. ME is supported by Kreitman High-tech scholarship.

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
