[Reviews · NeurIPS 2020]

Review 1

Summary and Contributions: This work proposes a new formulation of operators used in graph convolutional networks. The formulation is based on ideas from PDEs. The authors introduced the formulations for graph convolution, graph pooling, and graph unpooling. The authors experimented the proposed model on 3D classification and segmentation benchmarks. The experimental results show the model achieves comparable results with other compared methods.

Strengths: + The idea of using differential operators in graph CNN seems new. + The experimental results are comparable to other state-of-the-art methods.

Weaknesses: - The derivation of the formulation is very hard to follow. It is unclear what is the input and output of the operators. Equations seem to be disconnected. Symbols just appear out of nowhere. I recommend some polishing on the derivations. - One important claim of the DifGCN ops are its efficiency. However, only FLOPs numbers are compared. To make the case for efficiency, the authors could add some comparison on wall-time cost between different methods.

Correctness: I cannot check because I cannot follow the derivations.

Clarity: The experimental results part is clear. The methodology part is not very easy to follow.

Relation to Prior Work: Yes.

Reproducibility: No

Additional Feedback: I am not able to fully assess the technical correctness of the approach. Empirically, the proposed approach seems to achieve similar accuracy with other methods. I am sure the significance of this work merits acceptance to NeuRIPs. ============ Added after rebuttal: I have read other reviewers' comments and the authors' feedback. I feel the paper has presented a novel method and the results are solid. I updated my rating to reflect it. However, I strongly recommend the authors to improve the quality of writing for others to easily comprehend the idea.


Review 2

Summary and Contributions: This paper presented a novel graph convolution kernel based on discretized differential operators, along with pooling and unpooling operators to discretized differential operators defined on a graph for graph Neural Networks.

Strengths: -The differential operators defined on graphs can be useful for many applications of GNNs. -Performance of proposed method is superior to state of the art.

Weaknesses: -

Correctness: yes

Clarity: yes

Relation to Prior Work: yes

Reproducibility: Yes

Additional Feedback: UPDATE: After reading other reviews and author response I decided to keep my original score for this paper.


Review 3

Summary and Contributions: This paper proposes a parameterization of a spatial graph convolution based on differential operators such as gradient and Laplacian. Laplacian and gradient are then constructed on feature graph similar to discretized graph laplacians. Judging by ablation study, inclusion of Laplacian operator in the convolution kernel definition leads to superior performance over baseline spatial GCN that do not incorporate it on multiple benchmarks.

Strengths: The paper is overall well written and tackles important problem of defining a 3D convolution kernel which is very relevant to Neurips community (3D deep learning). The experimental validation is convincing as it outperforms baseline spatial GCN on multiple benchmarks. Ablation study is appreciated.

Weaknesses: --The pooling and unpooling operations, based on Galerkin coarsening, are very straightforward extension and also, not at all beneficial, as evident in Table 2 and 3 and 4 where including pooling either reduces the overall accuracy, Table 3 and 4 or at best increases the overall performance by .4 in Table 2. --The proposed method is really not close to state-of-the-art on segmentation benchmark,Table 4, as claimed in abstract. (57 against 64). -- I did not find any interpretation/related work for the Eq. 2 definition that uses Gradient and Laplacian to define the convolution kernel. --Ablation study is shown only on one benchmark. Authors should state/show if similar conclusion are drawn from other benchmarks too.

Correctness: Mostly except that the proposed method does not really obtain state-of-the-art results as claimed in the abstract. It outperforms GCN based baselines, including DGCNN, which is sufficient for experimental validation.

Clarity: Mostly except that I found the implementation detail using PyTorchGeometric very hard to understand.

Relation to Prior Work: Yes.

Reproducibility: Yes

Additional Feedback: While 3D deep learning is an important area for GCN, I am not sure why the motivation and experimental validation of DiffGCN is so centered around point cloud related 3D tasks. Surely, the construction of DiffGCN is generic enough to apply it to other classical GCN problems? If the laplacian term is behind the superior performance over spatial GCN baselines that do not include it, what about spectral GCN that are parameterized by graph laplacian? Post Rebuttal: Thank you for the feedback. I read the rebuttal and other reviews and my rating stays.


Review 4

Summary and Contributions: This work presents pooling and unpooling operations for 3D point clouds by formulating a graph convolution kernel based on gradients and Laplacian of the unstructured feature maps defined on a graph (see Eq.2). By then, the method defines "double message passing" between vertices via the connecting edges as (Section 3.1), and later adopts the Algebraic multigrid for pooling and unpooling (Section 3.2): - Adopt the Graclus clustering [31] to form aggregates (cluster of vertices in graph) and define the pooling operator, and - Adopt the smoothed aggregation [20] to define the unpooling operator. In the end, the paper presents three experiments: classification using ModelNet40, part segmentation using ShapeNet Parts, and semantic segmentation using S3DIS, as well as an ablation study. Overall, the approach helps reduce the network parameters needed in the graph convolutions, thus reducing the computational cost.

Strengths: The two interesting parts in this work are (i) the part on algebraic multigrid, which looks novel but too short, and (ii) the idea of using differential operators to formulate the graph convolution. Positive results were shown in the experiments: - For shape classification on ModelNet40, DiffGCN is found to performs better than DGCNN and LDGCNN, PointCNN, KCNet, etc. - For part segmentation on ShapeNet Parts, DiffGCN is found to performs better than DGCNN and LDGCNN, PointCNN, KCNet, etc.

Weaknesses: The ideas of Laplacian coordinates and differential pooling have been explored in existing works on graph neural networks, e.g., in those works on Spectral-based Convolutional GNN (see [1] below). So technically, in Section 3.3, can you provide comparison not only with standard CNN but also the recent graph neural network models and state the novelty of this work. The idea of adopting AMG is novel and the only work that I am aware of is [29] "Learning Algebraic Multigrid Using Graph Neural Networks," which is recently published in ICML 2020 (please update the reference). However, seeing Section 3.3, the contribution of adopting AMG is not very strong. This work can be stronger, if it explores AMG in greater depth. More importantly, since this work already cited [29], can you provide more explicit comparison with [29] at the end of the related work section (Section 2)? What is the technical difference between the two? Any advantage of this work over [29]? Also, I guess this submission could be treated as kind of concurrent with [29] (?), since ICML 2020 was held just recently. For shape classification on ModelNet40 and part segmentation using ShapeNet Parts, how about other spectral-based method? Also, DiffPool and RS-CNN? For semantic segmentation on S3DIS, many existing methods can perform much better, e.g., JSENet (67.7), KPConv (67.1), PointWeb (66.7), etc.

Correctness: See below for the details

Clarity: Mostly clear

Relation to Prior Work: Mostly

Reproducibility: Yes

Additional Feedback: Sec 1: Please use Section instead of section in the last paragraph of Sec 1, etc. Eq.(2): as compared with general convolution, is it better to write it as an approximation rather than as an equality? P.3 Line 105: what is stencil? Please define it, since the term "stencil convolution operator" is not well defined. Eq.(13): please define A Table 1: PoinetNet should be PointNet Table 3: you may bold to highlight the top-performing data value in each column Section 4.2: for semantic segmentation on S3DIS, we may do the experiment using 6-folds cross validation or just evaluation on Area 5, I guess you did "6-folds cross validation", please specify in paper. Table 5: what is pixel accuracy? Some of the typos: P.4: w.r.t P.5: leaner? P.5: ODEs,PDEs // missing space P.6: point-cloud Missing reference, e.g., [1] A Comprehensive Survey on Graph Neural Networks

[Author Response · NeurIPS 2020]

We thank the 4 reviewers for their useful comments.

R1:

Re recommend some polishing on the derivations : We separated our derivation two fold: one is a theoretical derivation that defines what are the differential operators. The second part elaborates on obtaining these operators using the popular package PyTorch-Geometric. In light of this remark we will improve the text in the final version.

Re the addition of comparison of wall-time between different methods: We agree. In addition to the FLOPs count we presented, we now measured the time for each block, which agrees with the FLOPs count. For instance, a forward pass of a DiffGCN block (ours) takes 58ms, while for DGCNN [24] it takes 121ms (on a Titan RTX GPU). This is in congruence with the results in Table 1, where DiffGCN and DGCNN require 61.3M and 167.78M FLOPs, respectively.

R2: We thank you for the appreciation.

R3 + R4:

Re the proposed method is not state-of-the-art on the segmentation benchmark in Table 4: We achieve SOTA on ShapeNet part-segmentation. Re S3DIS: we will rephrase the abstract to distinguish between the segmentation datasets in this aspect. We examined several papers (also following R4) and found that their results originate from different architectures, and most significantly larger networks in terms of parameters. For example KPConv used about 15M parameters while in the original submission we used 2.6M parameters. Despite the large time-consumption of these experiments, in the 6 days since the reviews were published, we managed to increase the overall 6-fold mIoU by 4%, from 56.9% to 60.9%. That is by using ResNet bottlenecks, as in KPConv, but still with about 2.6M parameters so that the training will finish on time. We will update our best result in the final version of this paper, if accepted.

Re Comparison with spectral approaches: Spectral convolutions are defined either by Fourier transform or polynomials of graph Laplacian matrix (GLM). The differences are as follows: 1) GLMs are combinatorial and not geometric - they are defined only by the connections between nodes. Shifting the nodes does not change the operator. Hence, the GLM is not a classical differential operator. 2) In our convolution we project the gradient and Laplacian, defined on an edge, to the 3 directions $x, y, z$. 3) Fourier transform of GLM is computationally expensive, non-local in space and learned for a specific graph-structure, i.e., it is not generalized for different graphs. 4) Using GLM polynomials, one can express the mass operator and the Laplacian (zero and first order polynomials). Higher orders result in a more spread convolution, and overall, the approach can only express symmetric matrices. The gradient operator is a non-symmetric operator, mostly used for edge detection, and cannot be expressed by spectral convolutions. As noted in [13], spectral methods lead to worse results than spatial - thus we omit it in our experimental evaluations.

R3:

Re Pooling and unpooling operations and their performance: Indeed the performance of our approach with and without pooling is comparable. However, pooling significantly improves the running time, FLOPs, and energy consumption of the network. This is highly important for various real-time applications, e.g. autonomous vehicles. Hence, pooling is an essential component to make GCNs practical.

Re interpretation/related work for Eq. (2) (defining conv kernel by Gradient and Laplacian): In section 3.3 we cite [17], showing an interpretation of convolution as gradient and Laplacian operators for structured CNNs. We also mention in this section that methods like DGCNN [24] can be seen as (non-directed) gradient methods, by nodal discretization.

Re Ablation study is shown only on one benchmark: Due to space considerations, and following [23] we showed an extensive study on one experiment. More ablation experiments will be added to the supplementary of the final version.

R4:

Re Comparison with AMG work at ICML20: Although it has a similar name, the purpose of this work is completely different than ours. It defines a network that builds AMG prolongation to solve the Poisson equation (that is, classical PDEs) on unstructured grids. It is not targeted at geometric deep learning tasks on point clouds for example.

Re Other methods for classification and part segmentation methods: Spectral methods are not effective for our needs as described in [13]. DiffPool [27] proposes pooling by learning a dense clustering matrix. This is undesired for geometric deep learning for two reasons: 1) It is impractical for graphs that include thousands of vertices. 2) Such learning is restricted to fixed input and output sizes. Our method does not suffer from these issues, and includes both novel convolution kernel and pooling methods. RS-CNN differs from our work in: 1) Our method utilizes feature derivative information, contrary to the euclidean distance in RS-CNN. 2) We operate on graphs and k-neighborhoods, while RS-CNN considers correlations between points in a spherical neighborhood in 3D. Overall, our method outperforms RS-CNN both on ModelNet40 classification and ShapeNet part-segmentation. DiffPool did not include these experiments.

Re Inaccuracies, suggestions and typos: We will correct the mentioned issues, and recheck the writing.

[Meta-Review · NeurIPS 2020]

The paper proposes a novel approach for a relevant problem and the differential operators are interesting/useful for the community. The paper lacks clarity and technical details should be better described in the camera ready version. After the rebuttal, the reviewers agree that the approach is sufficiently interesting, novel and performs comparable/superior to the state of the art and should be accepted.